# Electric Field Enhancing Artifacts as Precursors for Vacuum High-Voltage Breakdown

**Rolf Behling**

Royal Philips, 22335 Hamburg, Germany; Behling@wtnet.de

**Abstract:** Abrupt formation of plasma in a high-voltage insulating vacuum gap and subsequent discharge of electrodes limits the reliability of a class of vacuum electronic devices, such as X-ray tubes. It has been suggested that electron field emission from negatively charged electrodes would precede and initiate such discharge. Heating and evaporation of material upon field emission would cause dense plasma to develop in periods of nanoseconds. High-pressure plasma would expand from the cathode, eventually bridging the gap. Nevertheless, the very reason for the unredictable initial development of discharge events after long periods of reliable operation is still matter of debate. Experience from industrial processes suggests hydrocarbon contamination to degrade the electric stability of high-voltage gaps. While former attempts aimed at explaining high field emission by carbonaceous 2D structures or surface resonance effects, this paper discusses whether 3D structures may grow slowly, until their evaporation in a matter of nanoseconds. Similar to the production of carbon nanotubes, protruding structures might comprise carbon and, in addition, metallic nanoparticles, which would boost production of vapor during their explosion. The hypothesis was tested by scanning electron and energy-dispersive X-ray inspection of two cathodes of medical X-ray tubes, covered with metallic seed nanoparticles, which served as model systems. A third cleaner cathode was inspected for comparison. Although certain suggested conditions of carbon feed, elevated substrate temperature and nanoparticle contamination of the surfaces were met, images showed only a very weak sign of growth of suspicious carbon structures. It seems, therefore, unlikely that CNT-like structures are a major cause of high-voltage breakdown between electrodes of X-ray tubes.

**Keywords:** high voltage; X-ray; medical imaging; CNT; field emission; discharge; electrical stability; nanoparticles; work function; plasma

---

## 1. Introduction

Vacuum electronic devices, which operate under high voltage, such as X-ray tubes, may fail due to abrupt and unpredictable breakdown of the vacuum insulation. High-voltage breakdown may occur after thousands of seconds of reliable operation. Field emission (FE) from negatively charged electrodes is assumed one of the precursor processes for subsequent high-voltage discharge, e.g., in the form of explosive electron emission (EEE), see [1–5]. Under specific conditions, nanometer-sized field-emitting sites may start bearing a high current density, heat up, and generate vapor and later, high-density plasma near the cathode in a matter of nanoseconds. This process precedes build-up of microscopic dense clouds of hot plasma with a pressure on the order of $10^6$ Pa, melting of the cathode surface and high, mainly electron-driven, current in the gap. Figure 1 depicts the resulting artifacts in a medical X-ray tube after a series of such events. It shows the interior of the tube after a multitude of vacuum discharges during high-voltage conditioning. The anode disk was removed to allow access to the various structures. The picture shows the tube frame made from steel, chromium blackened at the interior surface, and the cathode head of the X-ray tube with two tungsten coil emitters mounted

on a circular holder. Micrometer-sized craters are visible on negatively charged surfaces like the cathode and in those areas of the tube envelope, which were located opposed to the (later removed) positively charged anode. Positive electrodes show very different artifacts. Thumbnail-sized regions of once rough but now smoothened black coating indicate high current flow in areas of a few square millimeters. This finding is in line with the EEE model [2,4]. It predicts cratering on the negative electrodes and high energy input to larger regions on the positive side. Electrons, which hit the positive electrodes, emerge from the plasma and are defocused by its convex surface during its expansion into the gap.

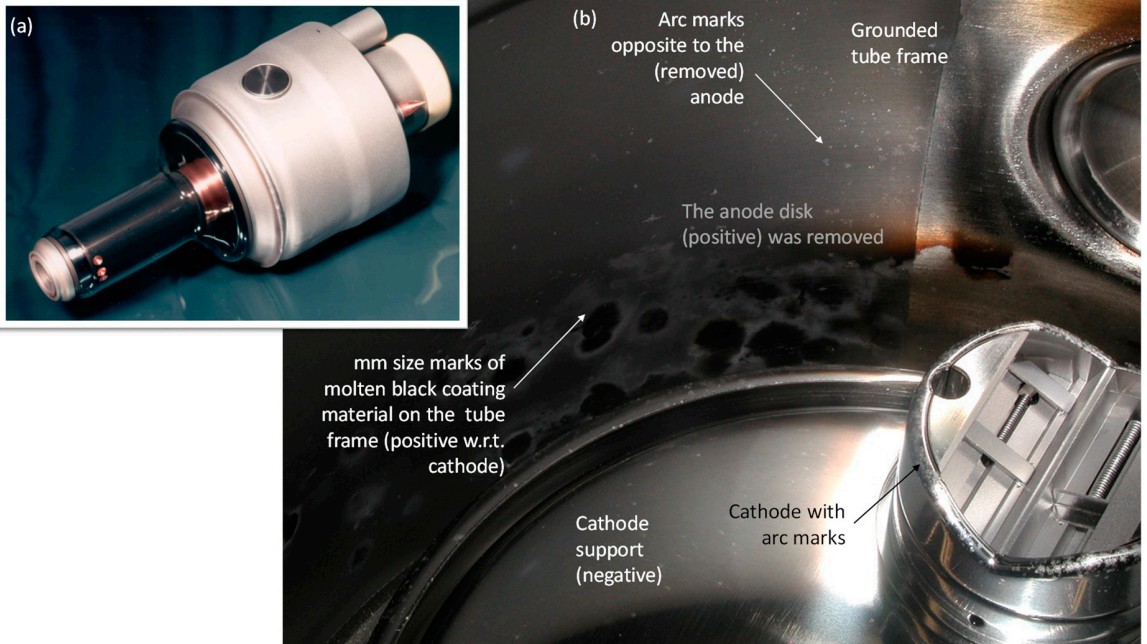

**Figure 1.** (**a**) Philips bipolar rotating anode X-ray tube for medical angiography, MRM 0508. The picture shows the 34 cm long tube insert, taken out of the tube housing. (**b**) Interior of the disassembled tube. The rotating anode disk was removed to allow for access. Sub-millimeter craters are visible as remnants of discharges on negatively charged surfaces like the cathode. The opposite black chrome surface of the tube frame shows molten areas of several millimeters in diameter. These were subject to electron bombardment from the expanding plasma during the development of discharges. Small craters are also visible on those black chrome surface areas, which were located opposite to the positively charged anode. Picture courtesy of Philips, see also [6], section 6.

In practice, the critical macroscopic electric field at the surface of negative electrodes is, by multiple orders of magnitude, too small to explain FE with sufficient current density from a clean and smooth metallic substrate. It is assumed, however, that microstructures like protrusions or other irregularities exist, which geometrically enhance the electric field or constitute local disturbances of the interface between metal and vacuum, which reduce the work function. The nature of such microscopic structures on negative electrodes, which enhance the macroscopic electric field to sufficiently high values for this process to happen, is still subject to debate. Assuming, that the work function is not substantially altered with respect to the rest of the substrate, Fowler–Nordheim (FN) voltage–current curves often suggest field enhancement factors (FEF) in stationary flow of FE between 30 and 1500. According to this understanding, FEF should be attainable from the slope of a $\ln(J/E^2)$ vs. $1/E$ graph, where J is the measured current from a FE center and E the macroscopic surface electric field. However, such artifacts with large FEF values are rarely observed in practice, e.g., by scanning electron microscope (SEM) imaging. One explanation for this lack of evidence of stationary structures may be their small size. Other reasons may be the poor contrast of carbon in an SEM or the temporal dynamics of their growth

and orientation. Vacuum breakdown often develops from an FE regime with currents on the order of microamperes to short circuit currents of kilo amperes in less than a microsecond for gaps on the order of several millimeters. Preceding processes may include a precursor phase where the field enhancing structure may form, as in the case of liquid electrodes. The initial phase of this process may last relatively long until FE current density suddenly exceeds a threshold and vapor emerges at a sufficiently high production rate. Once a cloud of vapor begins to develop about the source of field-emitted electrons, ion heating of the FE site would cause further enhancement of the electric field, which causes an avalanche of plasma production in a matter of nanoseconds. Figure 2 depicts the typical crater structure on a molybdenum cathode surface after a series of such discharges.

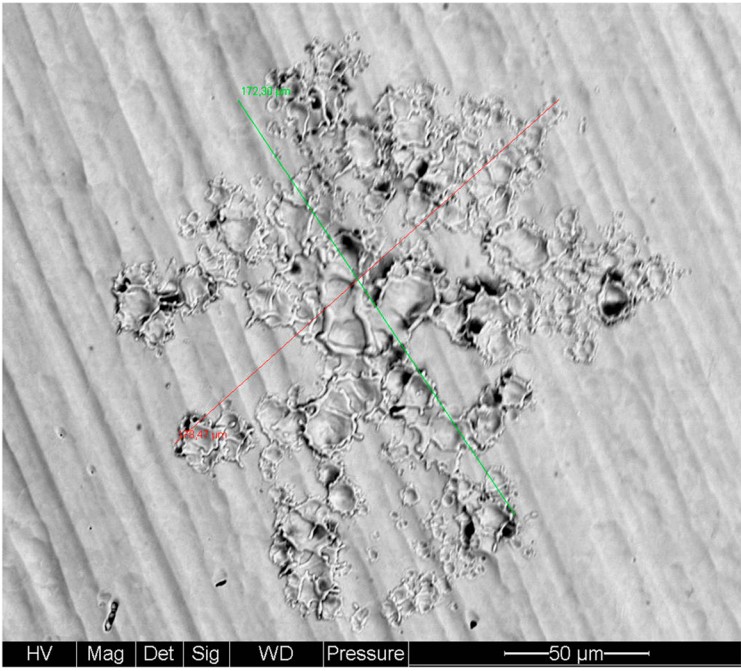

**Figure 2.** Crater structure on a negatively charged molybdenum electrode after multiple discharge events.

Experience from industrial production suggests the adverse role of carbon compounds, with a partial pressure beyond about $10^{-6}$ Pa in the residual gas (RG) atmosphere, on the high-voltage stability. The question arises of whether nanostructures like carbon nanotubes, graphene flakes or such artifacts may be responsible for high local field enhancement and subsequent breakdown. However, molecular dynamic modeling reveals that pure carbon nanotubes, possibly grown on an electrode surface, seem to bear too little material, or sublime too slowly. The structures may be sharp enough and conduct high field emission currents. However, they seem improper to deliver sufficient amounts of free carbon to efficiently feed a sufficiently dense plasma ball in an early stage of a discharge. Molecular dynamic simulation of the temporal development of heating and generation of vapor from sharp protrusions at the surface of a negatively charged electrode suggest that very specific conditions have to be met to initiate high-voltage vacuum breakdown, see [7,8]. Experience with carpets of carbon nanotubes which serve as field emission cathodes suggests, however, that these structures break down as well as other macroscopic smooth metal surfaces, see Figure 9b and [9]. On the other hand, stationary artifacts would have been removed already in the prior conditioning processes, where vacuum high-voltage discharges are triggered on purpose using overvoltage to destroy undesired weak spots on the electrodes. Conditioning of an X-ray tube may cause hundreds of individual discharge events and typically results in lowering the field emission and increasing the breakdown voltage. The question remains, why, despite prior conditioning, high-voltage discharges may occur abruptly again after hours or weeks of stable operation.

There is reason to believe that carbon-nanotubes (CNTs) would be preferred candidates for dynamically growing precursor artifacts. However, considering the discussion above, more aspects should be added to the theory. The mechanical robustness of CNTs allows for building highly field-enhancing structures with FEF values on the order of hundreds, see [9]. CNTs are used routinely as field-emitting cathode X-ray tubes [6]. However, as previously mentioned, MD modeling reveals that, e.g., single-wall CNTs (SWCNTs) on a metal substrate would not easily deliver sufficient vapor in sufficiently short periods of time upon heating by field emission and ion bombardment. They tend to sublime without igniting a plasma discharge in the vacuum gap. This would change if they would comprise more material and other material, e.g., inclusions of metal.

## 2. Investigated Model

In the following, we discuss the conditions for growth of electrically instable precursor artifacts with high FEF, based on carbon. Experience with the industrial growth of CNTs, the catalytic, nanoparticle-supported, radio frequency plasma-enhanced chemical vapor deposition (RF-PECVD), see Figure 3, may suggest to discuss a similar growth model for breakdown precursor structures.

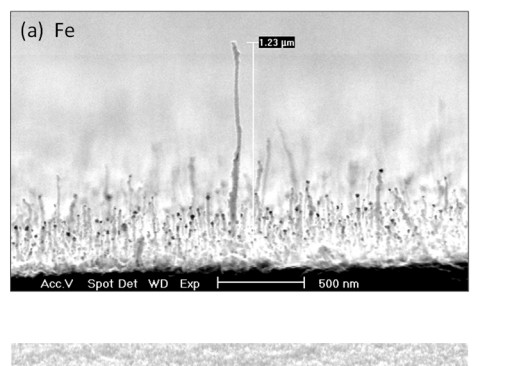
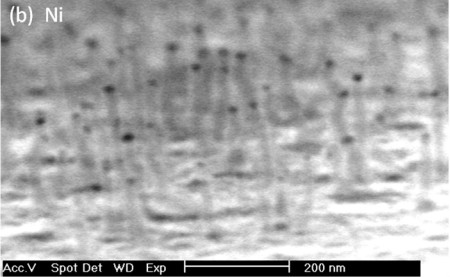
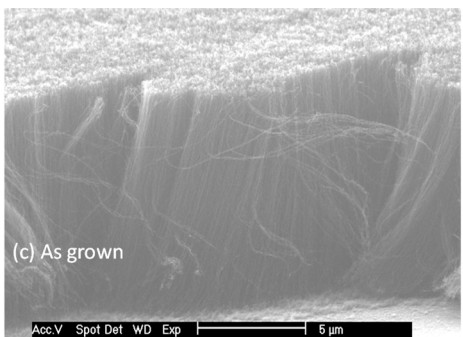
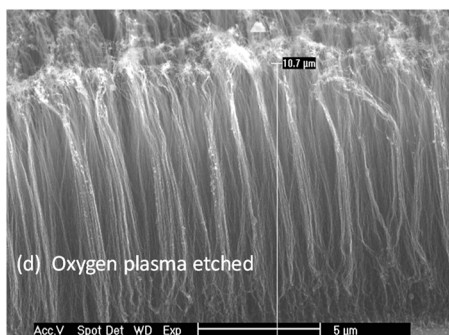

**Figure 3.** Metallic Fe or Ni nanoparticles at the apexes of RF-PEVCD grown carbon nanotubes. Growth paramerters: 13″ RF-PECVD system, 13.56 MHz RF input power 600W, gas stream $C_2H_2/H_2$ 8 sccm/40 sccm, substrate temperature 600–700 °C, pressure $10^{-3}$ mbar, 6 inch x 6 inch substrate, max -500V bias voltage. (**a**) Fe or (**b**) Ni as catalyst, Si substrate. (**c**) As-grown filamentary carbon artifacts. (**d**) Carbon-nanotubes (CNTs) after oxygen plasma etching. Picture courtesy of Philips.

### 2.1. CNT Growth Process

Under specific conditions, carbon diffusion through or along the surface of metallic nanoparticles results in the growth of carbonaceous fibers. CNTs may be harvested from the so-called vapor–solid–solid (VSS) growth mechanism under the conditions of RF-PECVD, see Figures 3–6 and [10–14] and the literature review [15]. Filament-like hybrid structures may comprise carbon fibers and metallic nanoparticles, typically grafted near their apexes. Depending on the conditions of catalysis, growth may occur according to this tip-model or, alternatively, by base growth, where the CNT stretches out from a nanoparticle which stays firmly attached to the substrate, see [16], Figure 11 and [17].

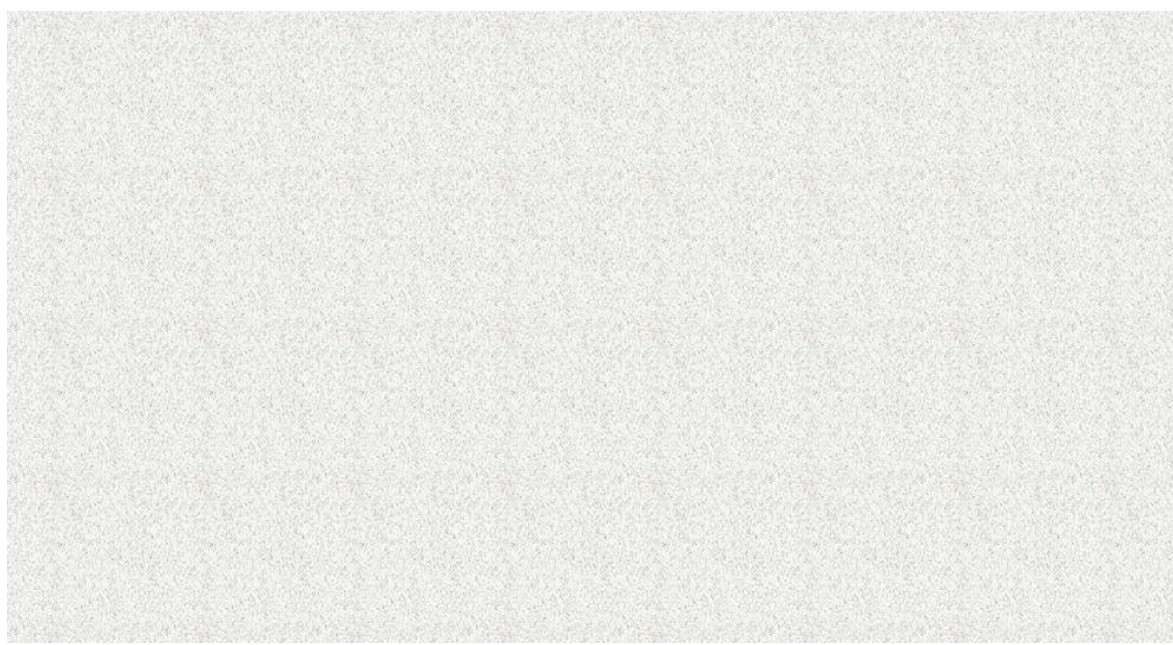

**Figure 4.** Left: PE-CVD reactor in operation. Right: bundles of CNTs, micro-structured by patches of catalytic layers (Pictures courtesy of Philips).

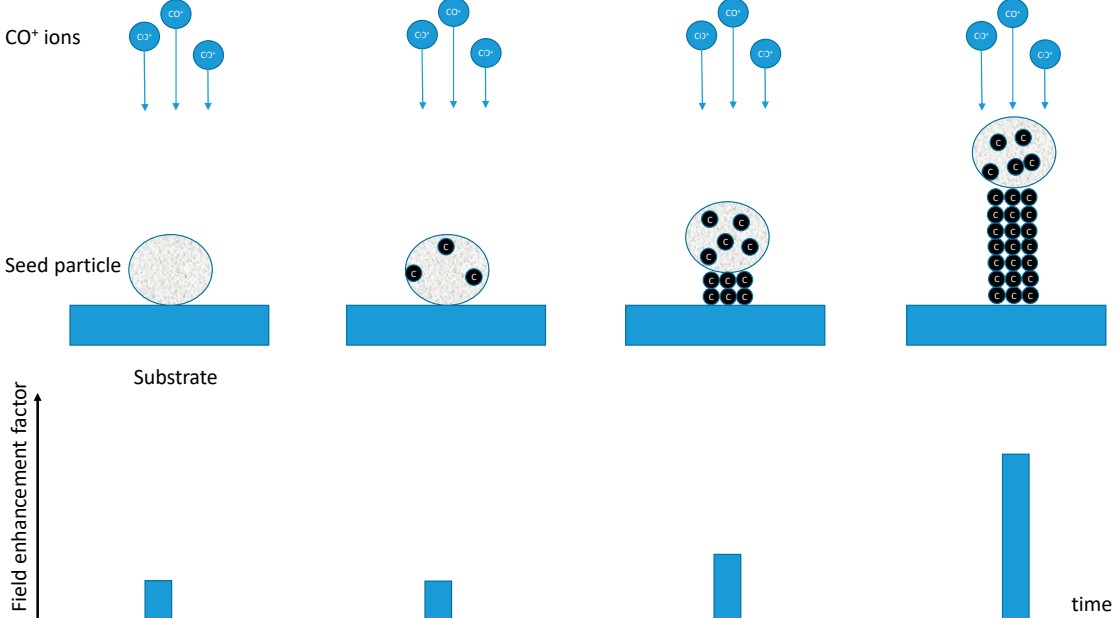

**Figure 5.** Top: Proposed model of tip-growth of breakdown precursors as metal containing CNTs by the vapor–solid–solid (VSS) growth mechanism. Carbon is fed by impact of CO$^+$ ions. Bottom: time sequence of the rising field enhancement factor. When a threshold field enhancement factors (FEF) is exceeded, high FE initiates the explosion of the tip.

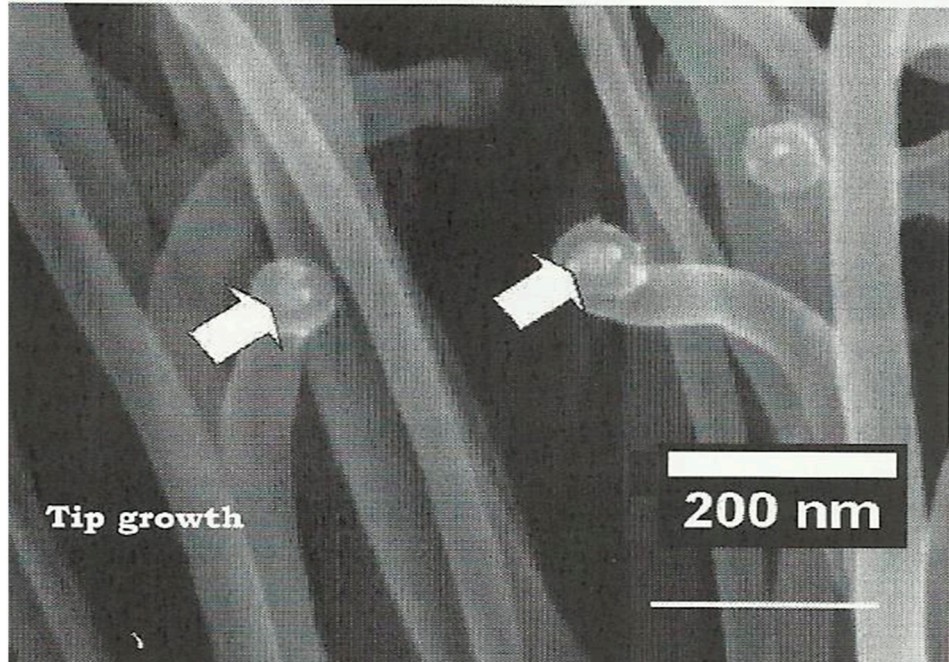

**Figure 6.** Copy of (Figure 8 in [10]), CNTs grown by the vapor–solid–solid (VSS) growth mechanism. Metallic particles are visible in the apexes of many CNTs.

Carbon compounds from the residual gas may crack at the surface of nanometer-sized metallic particles, which beforehand, were prepared on the substrate. In addition to the most commonly used hydrocarbons, carbon monoxide has also been known as a supply of carbon atoms for growth, notably of single-walled CNTs (SWCNT), see [17]. In an RF-PECVD process, carbon is supplied in the form of energized molecules and ions from a plasma discharge.

In preparation of the growth process, metallic catalyzing seed nanoparticles of a few dozen nanometers diameter are produced artificially, e.g., by laser ablation, surface melting of a nanometer-thin stack of metals, or they may be residuals of prior high-voltage discharges or other processes which create debris, see [4]. Preferably, they comprise metal with relatively large carbon solubility, such as the transition metals chromium, iron, nickel or cobalt or bi-metallic alloys. Despite its low solubility for carbon, copper may also act as catalyst [18]. Also, chromium and Ni–Cr [14], silver-, plutonium-, cadmium-, Zinc-, manganese-, molybdenum- [16], and tungsten-based bi-metallic particles were used [15]. Substrates used successfully in the past (cited after [15]) included silicon, silicon carbide, graphite, quartz, silica, alumina, magnesium oxide, calcium carbonate $CaCO_3$, zeolite and sodium chloride. This suggests that nanoparticles and substrates of a large variety of materials, as commonly used in vacuum electronic products, may be promoting the growth of thin carbon fibers. The microstructure of substrate and nanoparticles, such as the shape of crystalline steps, and the morphology of the interface affect growth rates.

According to the commonly accepted model of growth from gas, the surface of such nanoparticles catalyzes thermal cracking of adsorbing carbon compounds from the RG. For pure thermal catalysis, surfaces are often initially reduced by hydrogen treatment. Without bombardment of high-energy ions, temperatures above 400 °C, see [17], are required to achieve desired growth rates. Figure 3 in [17] shows a time-sequence of base growth of thin (<1 nm) SWCNT on cobalt nanoparticles, placed on a crystalline MgO substrate at 600 °C (accessed July 19, 2019, a video is available in the Supplementary Information of Reference [17]). While these SWCNT are semi-conductive, other SWCNT grown at 400 °C show different chirality and metallic conductivity.

It is suggested, however, that bombardment with ions of kinetic energies in the tens of keV range would also reduce and clean the surface of foreign layers, and deliver the energy required

for cracking and rapid diffusion even at low temperature. Atoms delivered by these ions deeply penetrate the seed nanoparticles. In RF-PECVD plasma reactors, the nanoparticles receive carbon on the plasma side which is exposed to reaction gases such as methane, acetylene or carbon monoxide [19]. After cracking, carbon atoms or dimers diffuse along the surface or directly penetrate the underlying nanoparticle and dissolve. In case of bulk diffusion, the carbon concentration may reach or exceed the solubility in the material at the exposed side. It is assumed that carbon atoms or molecules, which reach the meniscus between nanoparticle and substrate, precipitate and react with each other to form carbonaceous molecules like SWCNT or multiple-wall nanotubes (MWCNT), depending on the conditions which prevail, see [20]. During CNT tip-growth, which we primarily discuss in this context, the metallic nanoparticles must detach and be levitated from the substrate. This process and perhaps even the growth process may be further promoted in the presence of a pulling electric field. Eventually, metallic nanoparticles are embedded somewhere within the filament structure, see Figures 3, 6 and 7b. Pure CNTs will be produced by stripping off or etching away the metallic inclusions.

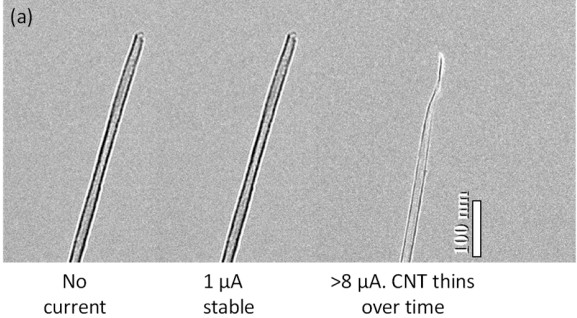

No current      1 µA stable      >8 µA. CNT thins over time

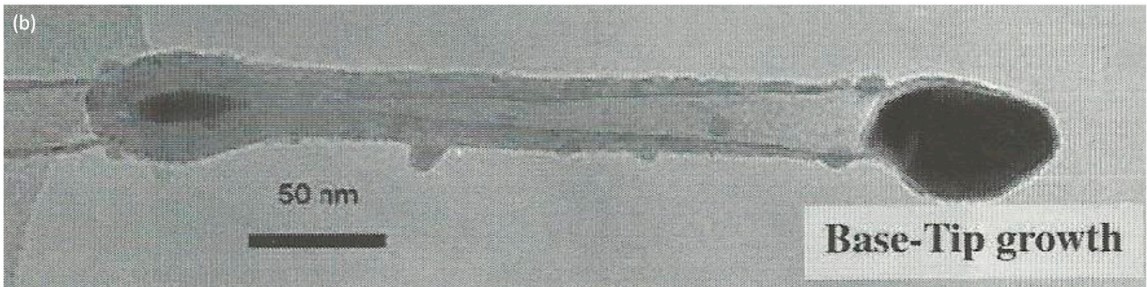

**Figure 7.** (**a**) Destruction of a CNT under FE (Picture courtesy of Philips). (**b**) Metallic particles in the apex and at the base of a CNT, which was grown by the vapor–solid–solid growth mechanism. Copy of (Figure 9 in [8]).

For comparison, Figure 7a depicts a transmission electron micrograph of a 300 nm long CNT without metallic inclusion, which had been subjected to FE. While it remains stable at currents below 1 µA, FE above 8 µA causes sublimation of carbon. Taking the FN plot, the typical work function for CVD-produced CNTs is derived as between 5.0 and 5.7 eV, see [21]. Under FE, the nanotube depicted in Figure 7a did not develop a plasma. Such behavior may change if a substantial amount of foreign material with high vapor pressure and reduced work function like metal is involved, as shown in Figure 7b. Of course, both terms, vapor pressure and work function, must be used carefully in this context of nanometer-sized apex structures of mixed material. They are used only to indicate the assumed tendency.

CNTs may also be grown between a sandwich of material, as shown in Figure 8 below, see also Reference [19]. The visible caps tend to shield the CNTs from destruction and even promote their growth. We may speculate that such macrostructures might also be responsible for the initiation of EEE or similar processes.

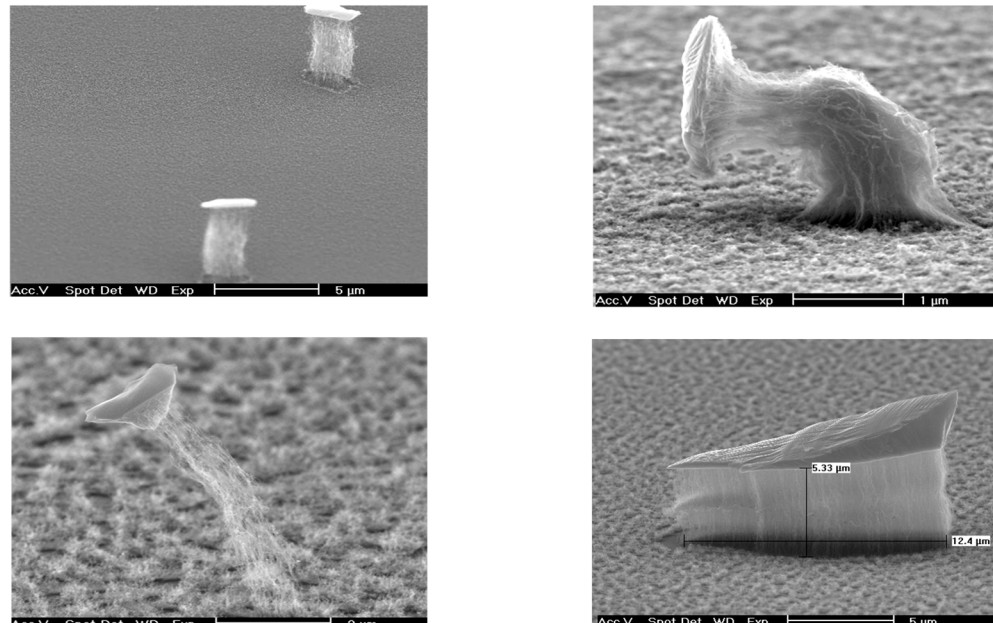

**Figure 8.** Sample pictures of CNTs grown by RF-PECVD in a sandwich structure between macro-particles, see [13]. Catalytic nano-particles are located within the individual fibers and feed carbon atoms into both ends of the growing structures. Pictures courtesy of Philips.

Other than for highly efficient field-emission cathodes, parallel alignment of the CNTs normal to the surface is not necessary for the initiation of the electrical breakdown of a macroscopic gap. Any filament structure, like the ones shown in Figure 9a, should enhance the FEF upon application of a high electric field. CNTs from ink-sprayed suspensions are erected by the initial application of high electric fields and subsequent fixation by baking for stable electron emission. The FE current rises due to a rising average FEF, see (Figure 2 in [22]).

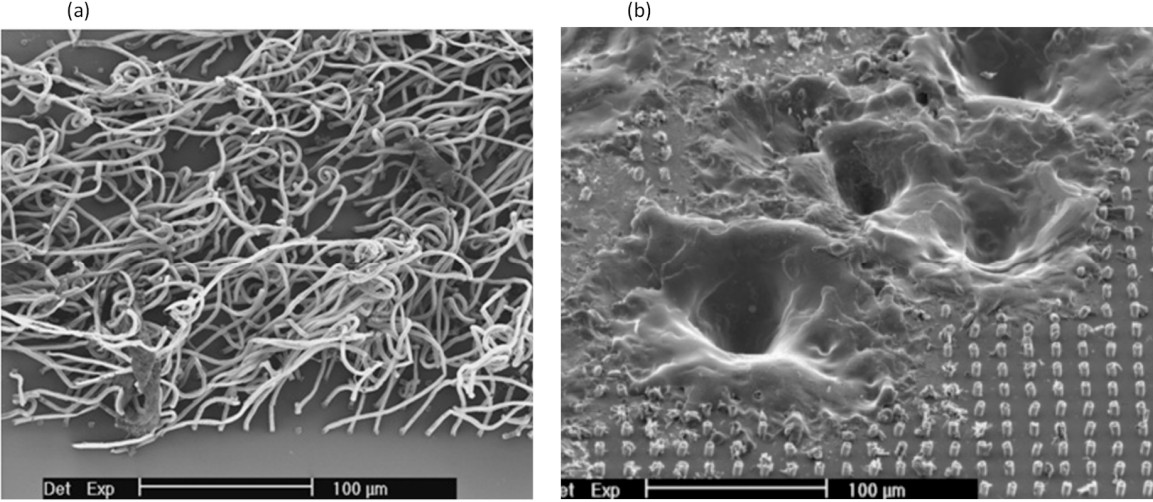

**Figure 9.** (**a**) CNTs grown under non-ideal conditions. (**b**) Cratering of the negative electrode coated with carpets of CNTs after application of excessive DC high voltage.

## 2.2. Nanoparticle Contamination

Nanoparticles with a broad distribution of size and surface morphology are typically present on extended technical surfaces. Despite cleaning, nanometer-sized clusters of chromium, iron, nickel, copper and alloys like stainless-steel may be constituents of residual particles in vacuum electronic

components. It has been observed, that under the stress of high electric fields during operation of an
X-ray tube and movements in X-ray systems, μm to mm sized particles may detach from their origin
and impact on negatively charged surfaces [3]. Typically, smaller particles stick by Van-der-Waals
forces unless they are large enough such that electrostatic forces exceed the sticking force, see [23,24].
As shown in Figure 10, steel balls larger than 100 μm in diameter tend to detach in practical macroscopic
electric fields of 10 kV/mm. Under real conditions, even smaller particles will detach. This may occur
when isolating contaminants cover the substrate, and when elongated particles and particles at the
apex of protrusions are involved. For illustration, Figure 10 shows in a comparison the result of a
sample calculation for a contaminated surface with one monolayer of oxide as model contaminant,
an elongated particle and a 5-fold field enhancement on top of a protrusion. Particles of a base diameter
smaller than 100 nm would stick to the surface. Therefore, the technical surface of a cathode electrode
is expected to be occupied by nanometer-sized particles even when an electric field on the above-stated
order of magnitude is present.

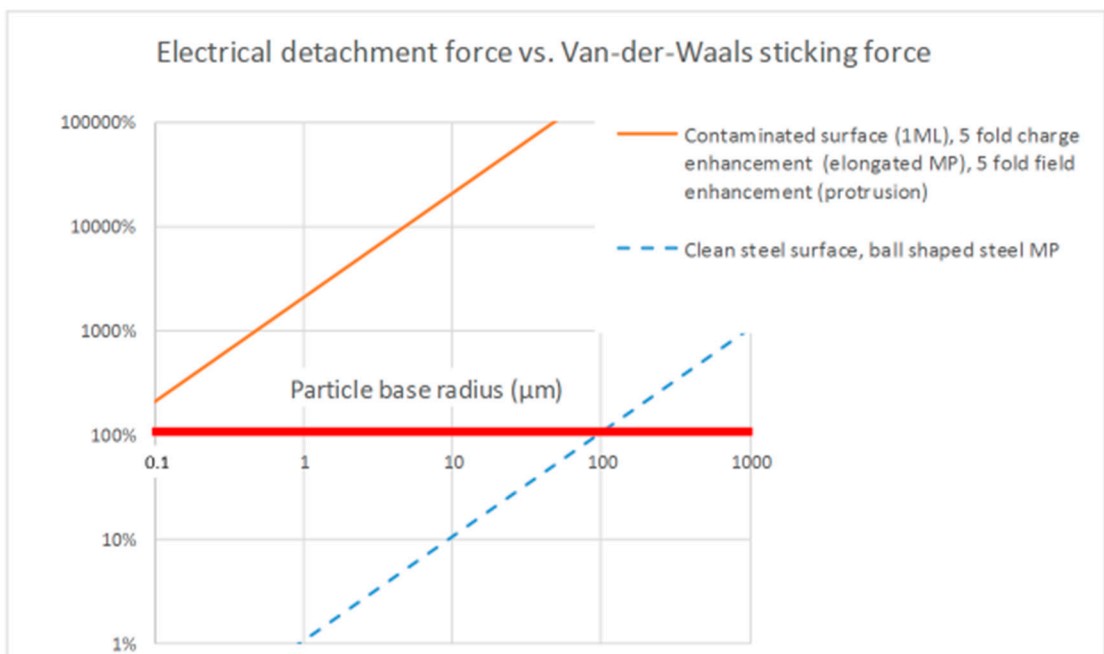

**Figure 10.** (Dashed line): Relation of the force required to detach chromium balls from a metallic
substrate and Van-der-Waals sticking force for different diameters of the balls for an electric field
of 10 kV/mm. The sticking force supersedes the electrical pull up to particle diameters of 100 μm.
(Solid line): Result of an example calculation for a contaminated surface with one monolayer (ML) of
oxide, a five-times elongated particle and a five-fold field enhancement on top of a protrusion.

Multiple authors have illuminated the influence of the surface condition and the size of catalyzing
nanoparticles for CNT growth, e.g., [25,26], differentiating base-growth from tip-growth. In the context
of high-voltage breakdown, tip-grown nanofibers should be more critical, as metallic material of
high-vapor pressure would be grafted in their apex. Tip-growth requires detachment of the nanoparticle
from the substrate. In order to achieve this, oxidized substrates, such as alumina, coated with thin
layers of catalyzer-material are often used to enable nucleation and the formation of non-aggregated
and detachable nanoparticles with a narrow distribution of size by annealing, see (Figure 4 in [26]).
Therefore, it is expected that non-metallic substrates, e.g., with oxidized surfaces, may promote
tip-growth in vacuum electronic products. In an X-ray tube, metallic vapor from various sources,
notably tungsten vapor and small atom clusters from the heated focal spot, may condense on negative
electrodes, nucleate and grow to build nanoparticles of a critical size on the order of a few dozen

nanometers. The state of oxidization, substrate temperature and other effects, which mobilize atoms, like ion impact, may, therefore, be of importance.

Another aspect to consider is the stability of nanostructures in the presence of a strong electric field. A rough assessment suggests that critical structures would stick under realistic assumptions. We assume that the sticking force of a carbon fiber would be on the same order of magnitude as the Van-der-Waals force of a chromium ball and that the same holds for the sticking force of a particle on top of a carbon fiber. Figure 10 then suggests that a 100 nm diameter chromium ball on top of a carbon fiber will stick and survive field enhancement of a factor $10^3$ before detachment. Substantial field emission by such structures, without detachment, seems realistic.

### 2.3. Carbon Feed

Hydrogen and carbon monoxide dominate the residual gas of most vacuum electronic devices like X-ray tubes. Typically, X-ray tubes are not actively pumped. Low residual gas base pressure is maintained by ion capturing. The anode of a rotating anode X-ray tube is a source of carbonaceous molecules, which will be adsorbed again upon cooling. During operation and heating, a typical titanium–zirconium–molybdenum (TZM) anode of a medical diagnostic rotating anode X-ray tube may produce up to $10^{-1}$ Pa CO and a smaller amount of $CO_2$, see [6] and Figure 11.

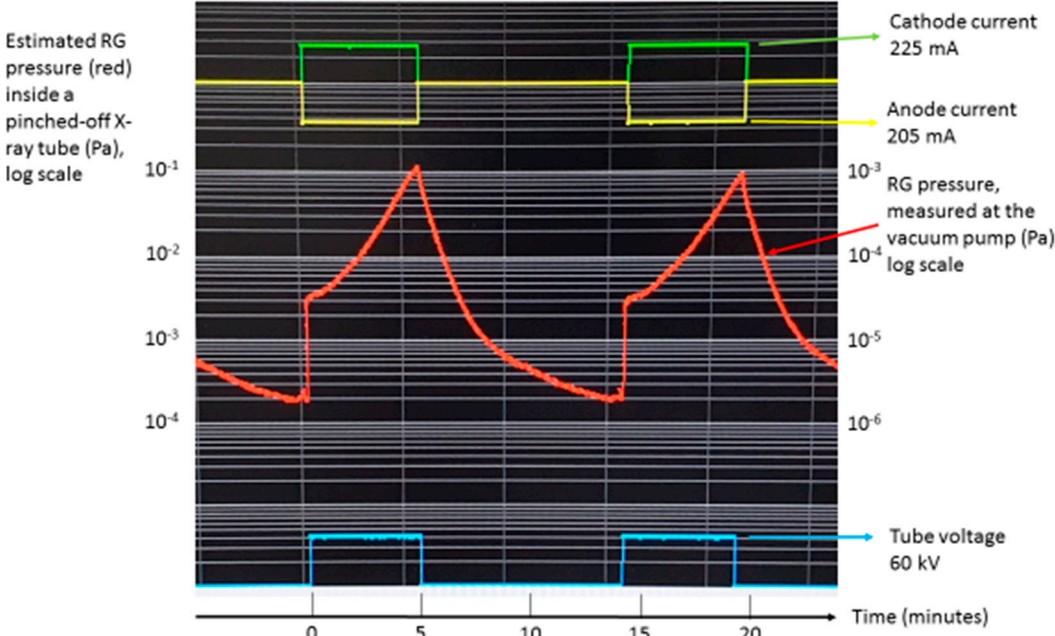

**Figure 11.** Residual gas pressure in a bipolar medical X-ray tube during exhaust, operated up to the highest permitted anode temperature. Load conditions for a 5 minute exposure time in each cycle are in this case: tube voltage 60 kV, cathode current 225 mA, anode current 205 mA. For a period of a few seconds, $10^{-1}$ Pa peak pressure is estimated for a pinched-off ion pumped tube.

Rotating anode X-ray tubes operate with electron currents on the order of up to about one ampere. The kinetic energy of electrons covers a range of up to 150 keV. Low-energy electrons in the primary beam from the cathode, and multiply backscattered electrons cause substantial ionization in the RG. Figure 12 schematically illustrates trajectories of ions, which, for this picture, are assumed to be generated only at the surface of the anode (which is not realistic in practice). Ions are buried in negatively charged electrodes. Ion loss by implantation reduces the RG pressure, and may yield pump speeds of on the order of 0.1 liters per second at full beam current. For almost all commercial X-ray tubes, this advantageous effect makes active pumping obsolete. However, the pressure is relatively high and may reach the Paschen limit of gas discharge in situations of thermal overload. Due to the

high atomic number and its high density, about half of the primary electrons backscatter from ordinary tungsten X-ray targets, where the X-rays are generated. Additionally, chemical getters help stabilizing the base RG pressure. In conjunction with ion bombardment of the cathode, $CO^+$ and $CO_2^+$ ions will be cracked. Typically, the focusing cup of the cathode operates at temperatures between 400 and 800 °C. These conditions, similar to the conditions of CNT growth in a RF-PECVD reactor, see above, may enable thermal cracking of neutrals and stimulate diffusion. As shown below, ion implantation further accelerates carbon diffusion. $CO^+$ ions may hit the seed nano-particles and cause a deep impact of carbon. It is, thus, expected, that the filamentary artifacts grow under ion impact even at lower substrate temperatures than stated above.

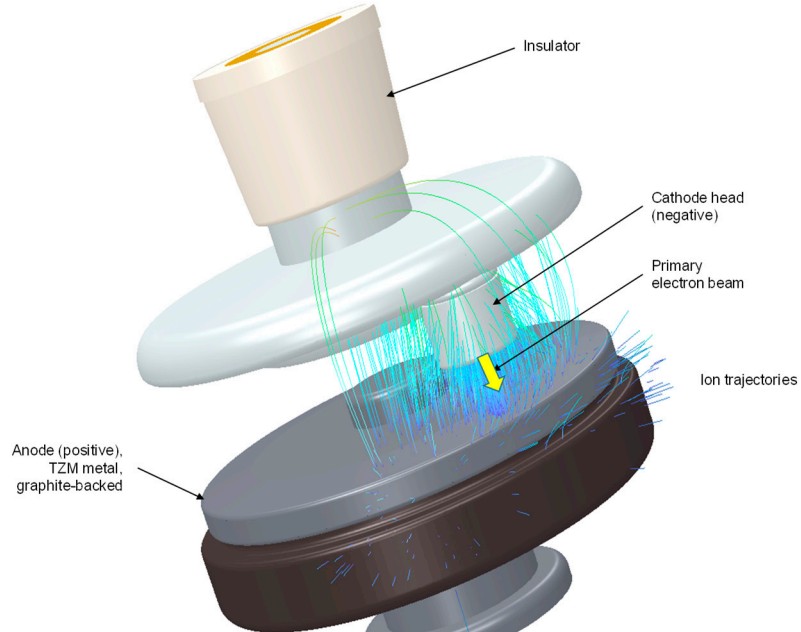

**Figure 12.** Illustration of a shower of ions generated near the anode surface from electrons which are backscattered from the focal spot of the X-ray tube after first impact, see Figure 14. Ions travel to the negatively charged cathode head and its supporting structure. Arrow: primary beam. Picture from a simulation: courtesy of Philips.

*2.4. Ion Impact Dynamics*

While low-energy ions may promote the growth of carbon filaments underneath, massive high-energy ions, which transit the nanoparticles, are expected to be destructive. Vacuum electronic components in which high-energy ions far beyond 150 keV dominate, should, therefore, not suffer from the mechanism described in this paper. The balance between sputter rate and carbon incorporation into nanoparticles, which depends on the ion impact energy, will determine whether a field-emitting structure may grow or not. Figure 8a may be indicative of shielding activity of macro-particles against ion bombardment. Heavy (Xe) ions with an energy on the order of 80 keV are known to destroy tens of nanometers large gold clusters, see [27]. Therefore, it is expected, that tungsten ions which emerge proximal to the focal spot on the anode of an X-ray tube will prevent build-up of hybrid CNTs. They typically impact at the center of the focusing structure of the cathode of an X-ray tube or directly hit the tungsten electron emitter coil with an energy of up to 150 keV. In practice, the electron emitter is very rarely seen to be the origin of EEE, see Figure 13. It is subject to direct ion bombardment from the focal spot. The lack of craters can also be interpreted as result of its high operating temperature of between 1800 and 2500 °C, see [6], and tungsten carbonization. This fact further supports the proposed model of initiation of EEE.

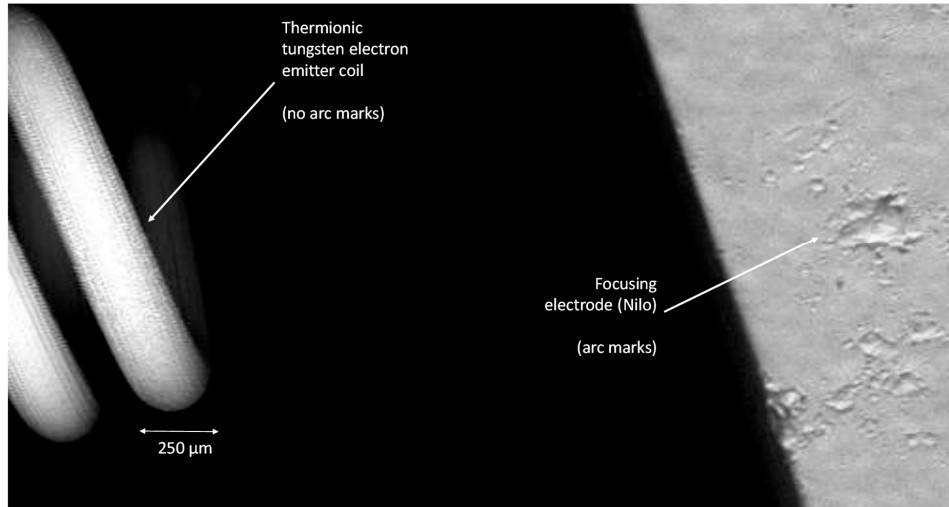

**Figure 13.** Picture of the cathode of an X-ray tube after discharge events. Left: tungsten emitter coil, free of craters, right: focusing electrode with a few craters. Dimensions: The wire diameter is 250 μm. The electron emitter is subject of direct high-energy heavy-ion bombardment from the focal spot (direct view), the focusing electrode to the right is not.

The following analysis will demonstrate the principle feasibility of the proposed growth model. The complexity of surface textures, surface chemistry, nanoparticle and ion dynamics and so forth would not allow for an accurate prediction of the time-to-discharge. Rough estimates of the dominant parameters must suffice. The RG pressure in a rotating anode X-ray tube may exceed $10^{-3}$ Pa for about up to a minute during operation, i.e., exposure of a patient. It may peak even beyond $10^{-2}$ Pa for a few seconds at the end of a computed tomography scan. Consequently, ions are generated through the interaction of gas molecules and electrons. Ions penetrate all negatively charged electrodes. Figure 14 depicts the basic geometry. The ionization cross-section for backscattered electrons is highest close to the cathode, where their kinetic electron energy reaches its minimum. For simplicity, the residual gas pressure for the following calculation is approximated by an average for the time of operation. Ionization cross-sections shown in Figure 15 are taken from NIST [28]. The total electron backscatter ratio of tungsten is assumed to be 48%, the dimension of the bombarded area on the cathode is assumed to be 2 cm$^2$.

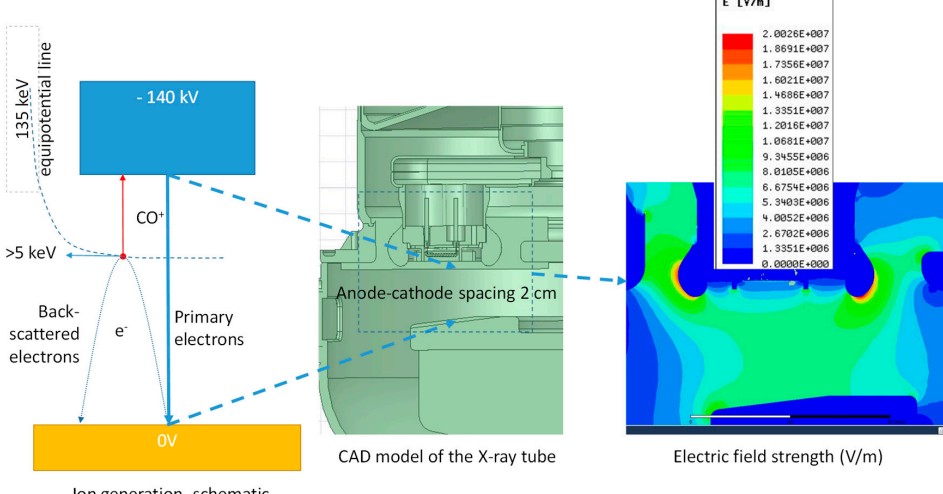

**Figure 14.** Sample trajectory of backscattered electrons and a typical point of ionization (left). CAD model of the X-ray tube in question (center). Simulated electrical field strength (right).

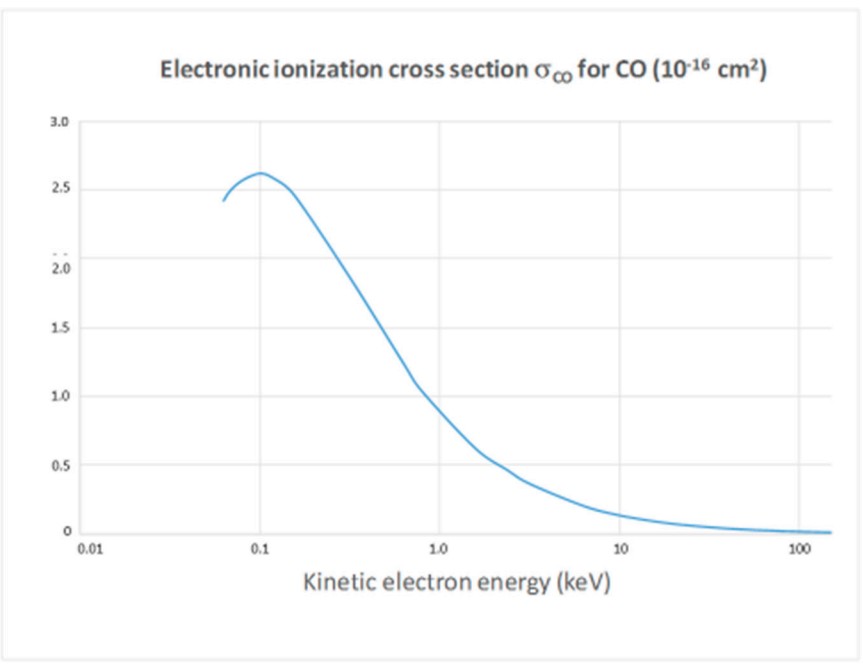

**Figure 15.** Ionization cross-section of carbon monoxide, see [18].

Figure 16 depicts the result of a simulation of the penetration depth of ions in stainless steel, which is a representative of the material of seed particles (simulation tool SRIM-2008, see http://www.srim.org, accessed on 16 June 2019). Carbon monoxide ions are assumed to decompose upon impact, see [29]. The impact energy would split between oxygen and carbon atoms, according to their atomic masses. The resulting penetration depth of the carbon atoms alone from decomposed ionized molecules is shown as bold line to be below 80 nm even for the most energetic particles, which appear in a medical X-ray tube. Figure 17 shows the calculated spectrum of ions, which are generated in the cloud of backscattered electrons. It is assumed, that the dominant portion of ions stick to the seed nanoparticles, and that high-energy ions would not destroy them or the underlying carbon fibers. The spectrum was simulated under the assumption of a rotational symmetric distribution of scattered electrons with a cosine angular distribution about the normal and a homogeneous electric field between anode and cathode. The distribution shows the ion flux integrated over the circular area of 2 cm² of the bombarded cathode head. Given the high ionization cross-section close to the cathode, where the backscattered electrons are slow, and the dominance of ions with energies around 30 keV, realistic depths of penetration of carbon ions are in the range of 20 nm. Therefore, the vast majority of carbon atoms will stick in the chromium seed particles in question. The same holds for nickel, iron and similar metals and compounds. For more details on this material combination, see, e.g., [30].

Primary electrons also contribute to ionization. However, such ions typically land opposite to the focal spot of the X-ray tube on surfaces in the middle of the cathode where the macroscopic electric field is typically low. Compared with the ions from backscattered electrons, it is expected that a large portion of the ions originating from the focal spot are "destructive" tungsten ions with a kinetic energy close to tube voltage (up to 150 kV) times elementary charge. They tend to destroy seed nano particles and filamentary structures with high FEF. Therefore, they are ignored in this discussion.

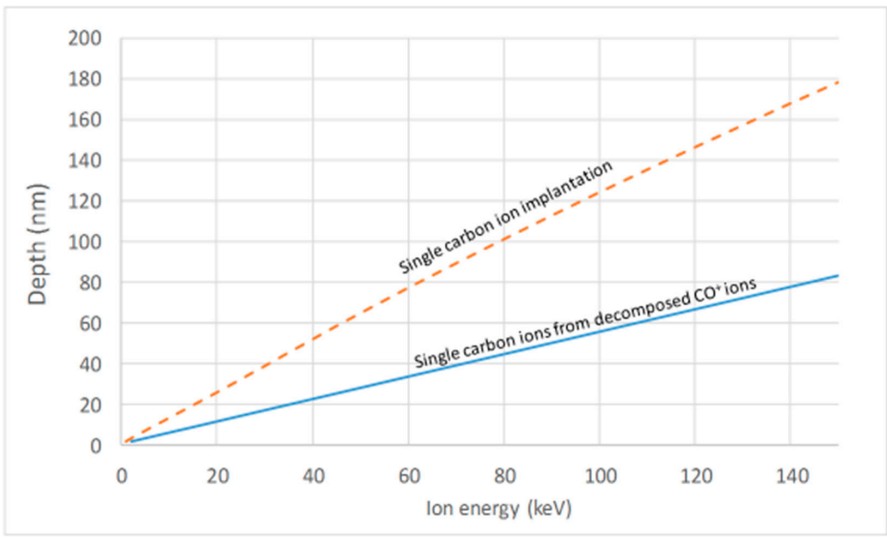

**Figure 16.** Simulated range of carbon ions in a sheet of chromium.

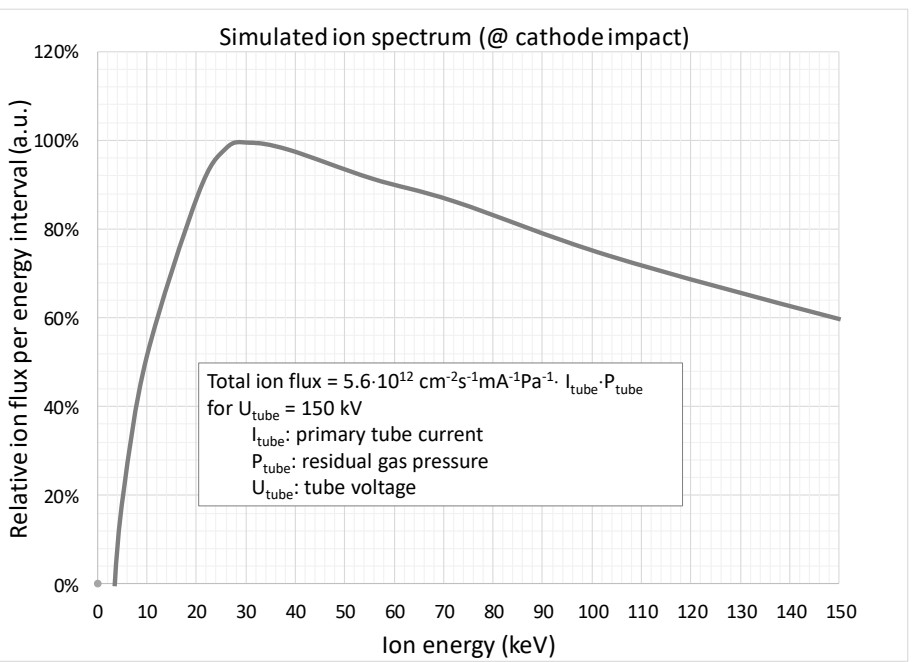

**Figure 17.** Calculated impact energies of $CO^+$ ions on the cathode rim, normalized to the maximum.

### 2.5. Time to Saturate

The characteristic time until breakdown according to the model to be tested, i.e., the time required for growth of a carbon fiber beneath a seed particle, is the time needed for saturation of such a particle and the time of subsequent elongation of the fiber. For typical operating conditions, the $CO^+$ ion flux at the outer rim of an X-ray tube cathode is on the order of $10^{12}$ ions $cm^{-2}$ $s^{-1}$ at an RG pressure of $10^{-3}$ Pa CO and a primary electron (tube) current of 500 mA. Starting from zero carbon content and assuming that all ions would stick after impact on a nanoparticle, the estimated atomic carbon solubility of 5% in the case of a chromium nanoparticles would be exceeded after a time, which depends on gas pressure, electron current and size of the particle. Its diffusion will result in in the appearance and segregation of carbon at the interface between nano particle and substrate. It is assumed, that the particles are held at elevated temperatures or that ions impact at such high energy that diffusion within the nanoparticle is not the limiting factor for growth of carbon filaments. Figure 18 shows the time required for a seed

nanoparticle to accumulate the maximal solvable carbon content. As there always exists some initial carbon in the nanoparticles, this time is regarded as a maximum characteristic time, which exceeds typical growth times, as mentioned in Section 3.

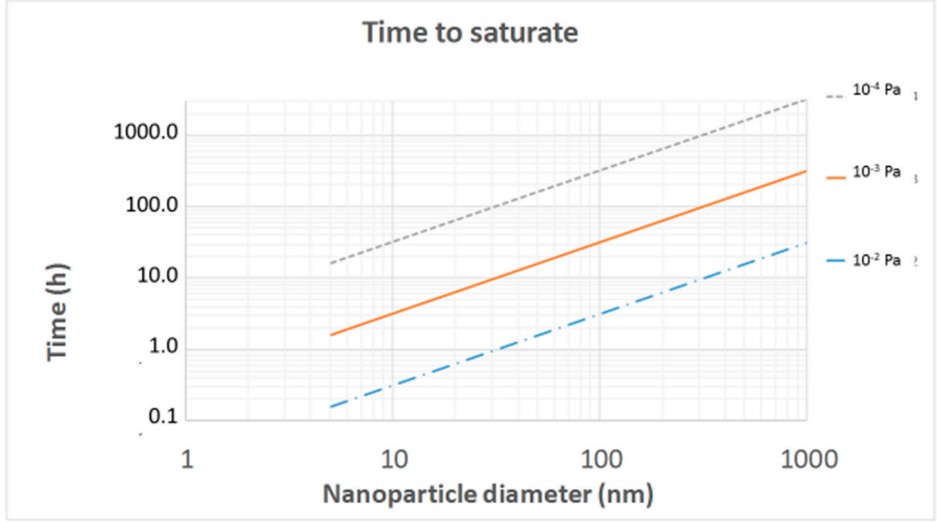

**Figure 18.** Hours to saturate ball-shaped chromium nanoclusters of diameter between 1 nm and 1 μm with carbon under the conditions stated in the text. Parameter: Partial RG pressure of CO.

If the substrate has lower carbon solubility than the seed nanoparticles, be it, e.g., molybdenum, back diffusion and segregation of carbon atoms from fast ions that may have transitioned the seed nanoparticles towards the interface would also contribute to an accumulation of carbon atoms at the interface.

*2.6. Stability of the Hybrid Structures Under High RG Pressure*

It is recommended for stable operation of FE cathodes with CNTs that the residual gas pressure of "ordinary atmosphere" (mainly CO and $H_2$) should be kept below $10^{-5}$ Pa to ensure stable operation in vacuum electronic components for on the order of $10^5$ seconds. However, it should be noted, that this pressure limit holds for pure pre-formed stationary MWCT and SWCNT field emitters intended for a long operating time, typically hours. The temperature at the apexes and in disturbed ("elbow") areas along the fibers with high electric resistance reaches values where oxidation and sublimation occur over a long period of time. The critical gas pressure for short-term destruction is expected to be orders of magnitude higher for the suggested hybrid CNTs with metallic apexes. They should dynamically change their length and FEF. Sublimation is reduced due to the larger radius at the apex. According to the FN theory, the FE current is steeply rising with the FEF shortly before EEE sets in. As shown in Figure 18, the characteristic time of artifact growth is on the order of $10^3$ seconds with a time of maximal current flow at high FEF, which is orders of magnitude smaller. If we would assume 1 second at highest current, the extrapolated critical gas pressure for destruction would be $10^5$ times larger than the recommended, i.e., 1 Pa. This RG pressure is beyond the Paschen gas discharge limit in a typical X-ray tube and is never reached in practice. Thus, the dynamically grown hybrid CNTs are expected to be stable in an ordinary non-oxidizing gas atmosphere of a real X-ray tube as long as FE is still limited. The situation is expected to change for highly oxidizing residual gas with large water or $O_2$ content.

## 3. Probability of Filament Growth under Non-Ideal Conditions

One may argue against the proposed model that carpets of CNTs for field emitting cathodes can only be produced under very well defined conditions and in fine-tuned processes. CNTs intended for cathodes should grow only at specified sites in the cathode. The fibers should be densely packed,

aligned normal to the surface, have isotropic length, good adhesion to the substrate, low electrical internal resistance and at the interface with the substrate. The ranges of suitable growth parameters are relatively narrow in order to achieve perfect field emitters. Experience indeed tells that the growth process for such emitters is very sensitive to the characteristics of the substrate and the size, surface condition and material of the catalyzer material. Also, gas composition, pressure, power applied for the feeding plasma discharge, bias voltage of the substrate, substrate temperature, etc., influence the result substantially.

An optimal process typically delivers CNTs of a density on the order of $10^5$ cm$^{-2}$ in about $10^3$ s. On the other hand, it would be sufficient to render an X-ray tube unreliable with respect to electrical discharge if the carbon fiber density would be on the order of 0.1 to 10 cm$^{-2}$. Given a typical discharge frequency in practice of about $10^{-4}$ Hz, fibers may be produced slowly. Thus, given the same density of nanoparticles, it would be sufficient to grow at a 10 times lower speed and a $10^4$ to $10^6$ fold reduced spatial density. Thus, "sub-optimal" conditions may still yield precursor artifacts.

## 4. Initiation of EEE

During field emission and heating beyond the temperature of the substrate, the pressure of the metal vapor generated from hybrid fibers, is expected to be more than two orders of magnitude higher than for pure CNTs, see Figure 19. Thermodynamic modeling of a metallic protrusion suggests the development of excessive temperatures inside the volume in comparison to its surface, see (Figure 1 in [4]), and consequently, high thermo-elastic stress in the peripheral structure. Such a burst may accelerate catastrophic release of material in the form of steam and liquid into the vicinity. In a different scenario, a metallic inclusion at the apex of a carbon nanostructure may simply melt and form a sharp tip under the action of the electric field by a Tonks–Frenkel mechanism, see [31]. This would further enhance field emission and accelerate the avalanche of destruction. Suggested by Figures 6 and 7, and as discussed in [4], droplets, carrying a load of metal, may also detach from hybrid tips after overheating the bottleneck beneath the apex of the hybrid structure. The electric field between droplets and residual apex would cause a high electric field and enhanced FE. The droplets may evaporate due to electron and ion heating. Geometrical disturbance of a hybrid emitter structure may be a key factor in the development and maintenance of EEE. Future research may test this hypothesis, e.g., by molecular dynamic modeling.

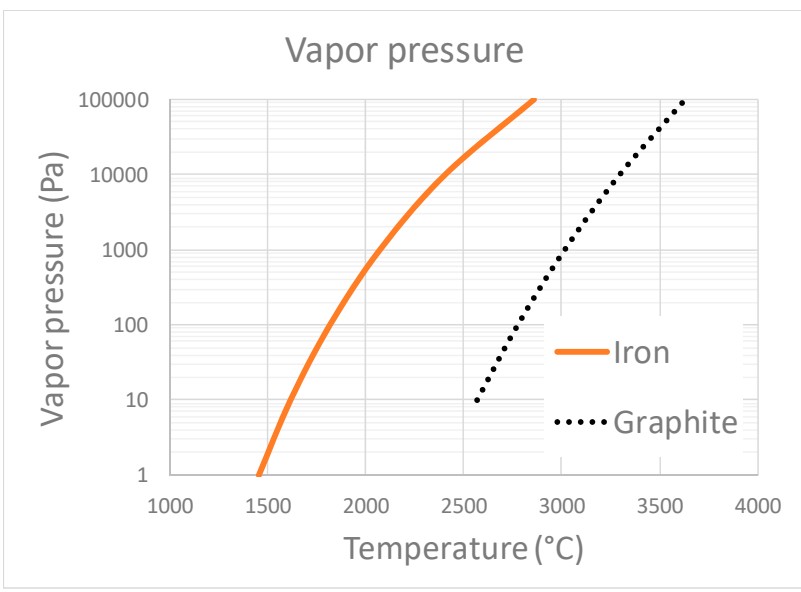

**Figure 19.** Temperature dependence of the vapor pressure of carbon (dotted line) and iron (as a model for other metals, bold line).

## 5. SEM Imaging

Nanometer-sized in-situ tip-grown filaments should be detectable in a high-resolution scanning electron microscope (SEM). They should dwell primarily under the "hood" of metallic nanoparticles. In order to test the proposed model, three sample cathodes from dissected medical X-ray tubes were thoroughly inspected with SEM and energy-dispersive X-ray analysis (EDX), after they had experienced high-voltage breakdowns during operation, see Figure 20. The two tubes, discussed in Section 5.1 had a molybdenum cathode ring exposed to high electric field and metal center section tube frame as depicted in Figure 14. The other one, discussed in Section 5.2, served for comparison and was from a glass tube with Ni42 steel cathode.

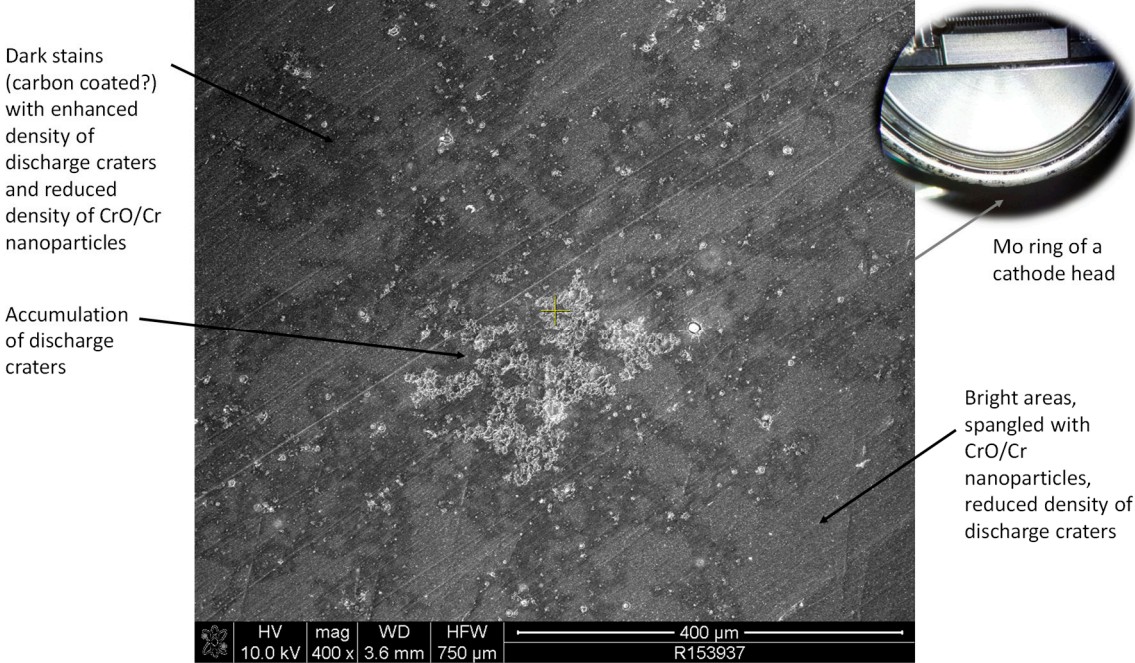

**Figure 20.** SEM inspection of a cathode Mo ring.

### 5.1. Mo Cathode Rings, Contaminated with CrO/Cr Nanoparticles

Two sample tubes were equipped with a black chrome coating layer, which released nanoparticles. Thus, the chance of discovery of carbon filament growth was greatly enhanced.

The outer molybdenum rings of two of the three cathode heads were exposed during conditioning to a macroscopic electric field strength of up to 25 kV mm$^{-1}$. The rings were contaminated with CrO/Cr nanoparticles in the form of droplets and other debris, as shown in Figures 21 and 22.

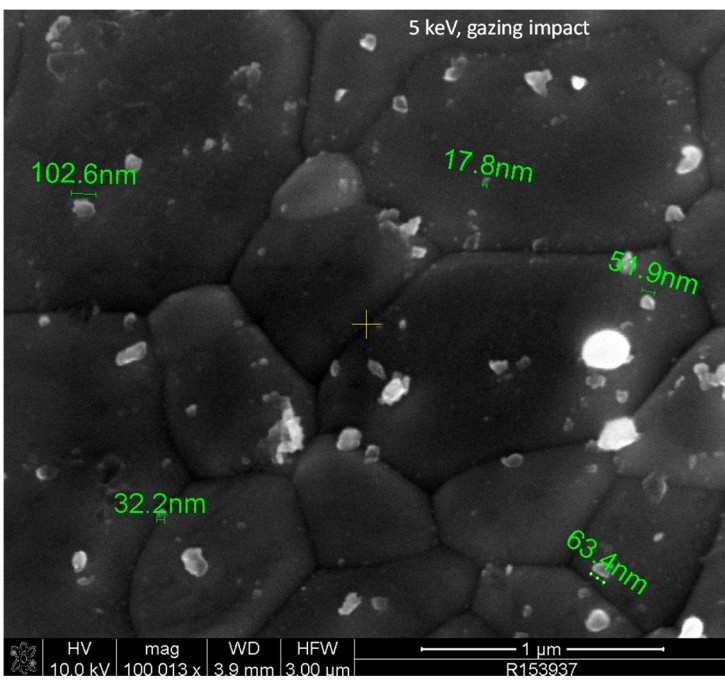

**Figure 21.** Chromium pollution on Mo ring of a cathode head and dimensions of nanoparticles.

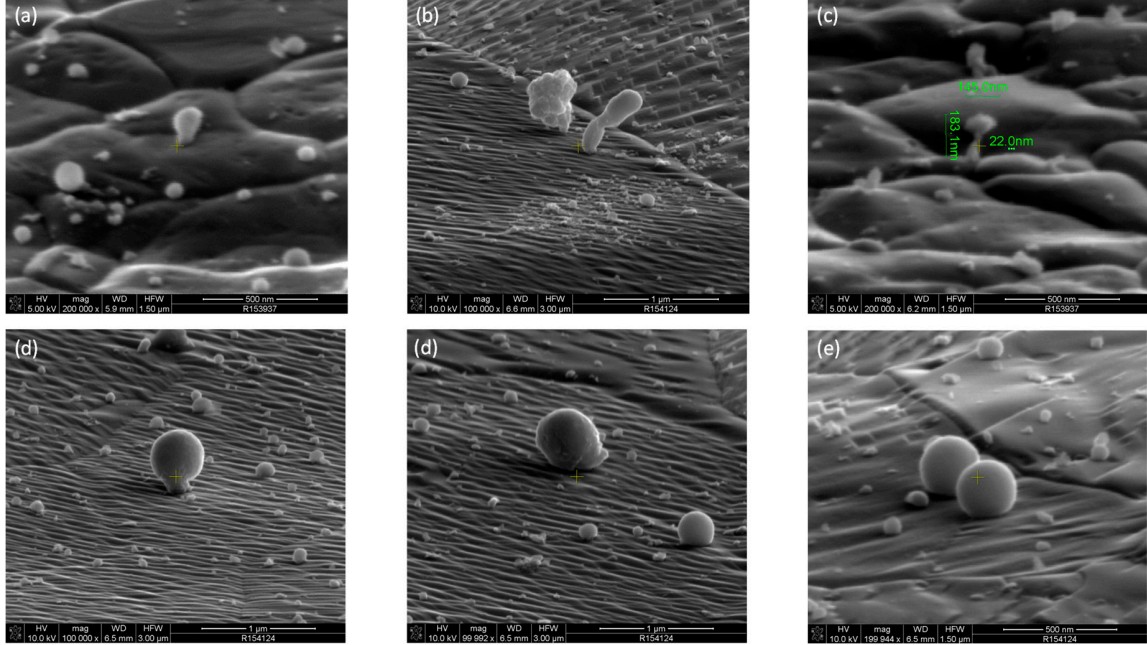

**Figure 22.** Selected artifacts of chromium pollution on a Mo shield of a cathode head. (**a**)–(**d**) Mushroom type; (**c**) with dimensions, see its EDX spectrum in Figure 27; (**d**)–(**f**) Ball-type artifacts.

The nanoparticles had obviously been transported by electrical force. Their exact composition could not be derived accurately from the EDX spectra. [32] reports a ratio of chromium metal/chromium oxide of 10/27 for the utilized galvanic process.

Figure 23 depicts a selection of discharge craters of various dimensions. It is obvious that small craters are far more dominant. There is no small crater found in any large one. The development of small craters seems to have preceded the creation of larger ones.

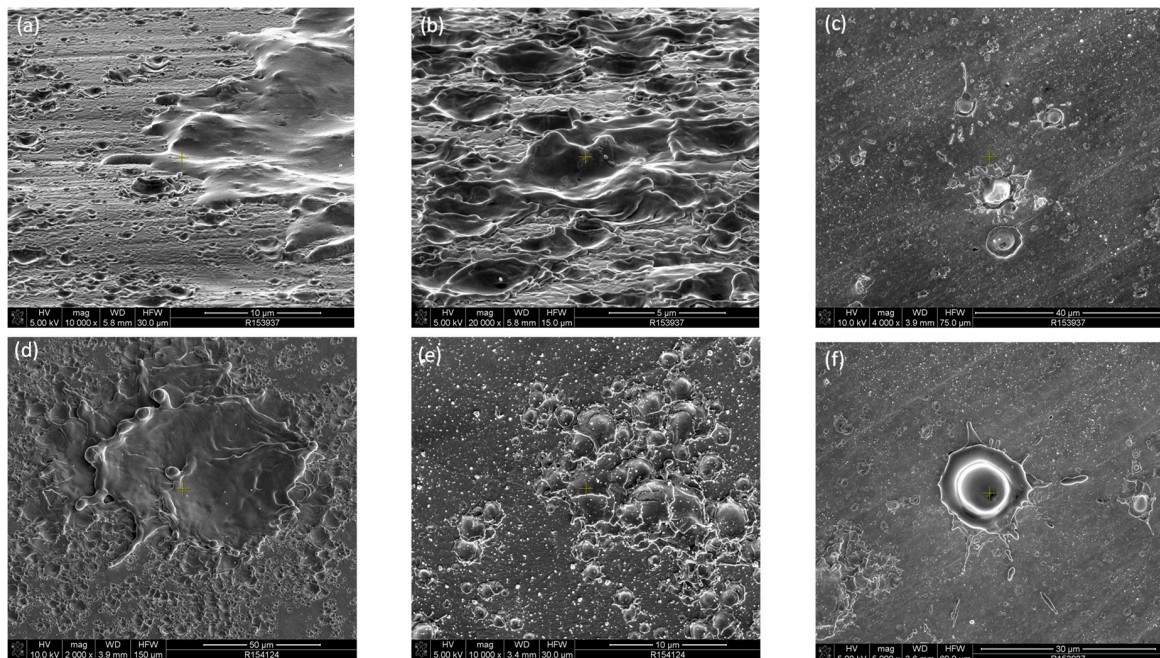

**Figure 23.** Discharge craters on a Mo shield after operation with a macroscopic electric field of max 26 kV mm$^{-1}$. (**a**) Sub-micrometer craters (which appeared first?), superimposed by large molten area; (**b**) Micro craters of different size; (**c**) Single craters, lower right crater later polluted with CrO/Cr nanoparticles (white dots); (**d**)–(**f**) Craters of very different size, similar to (**a**), later polluted with CrO/Cr nanoparticles (white dots). No sign of carbon fiber growth underneath the CrO/Cr particles.

Densities of craters and nanoparticles are anti-correlated, as shown in Figure 24.

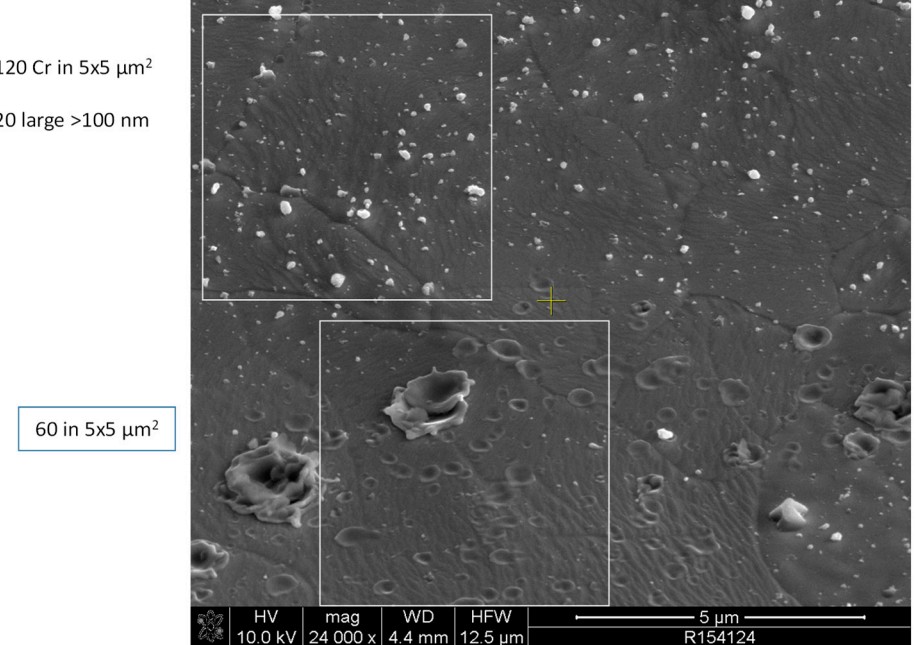

**Figure 24.** Density of discharge craters compared with the density of CrO/Cr nanoparticles. Densities are on the same order of magnitude, but surface patches with many arc marks are less populated with small nanoparticles and vice versa.

Figures 25 and 26 show the results of EDX inspection of one of the Mo cathode rings.

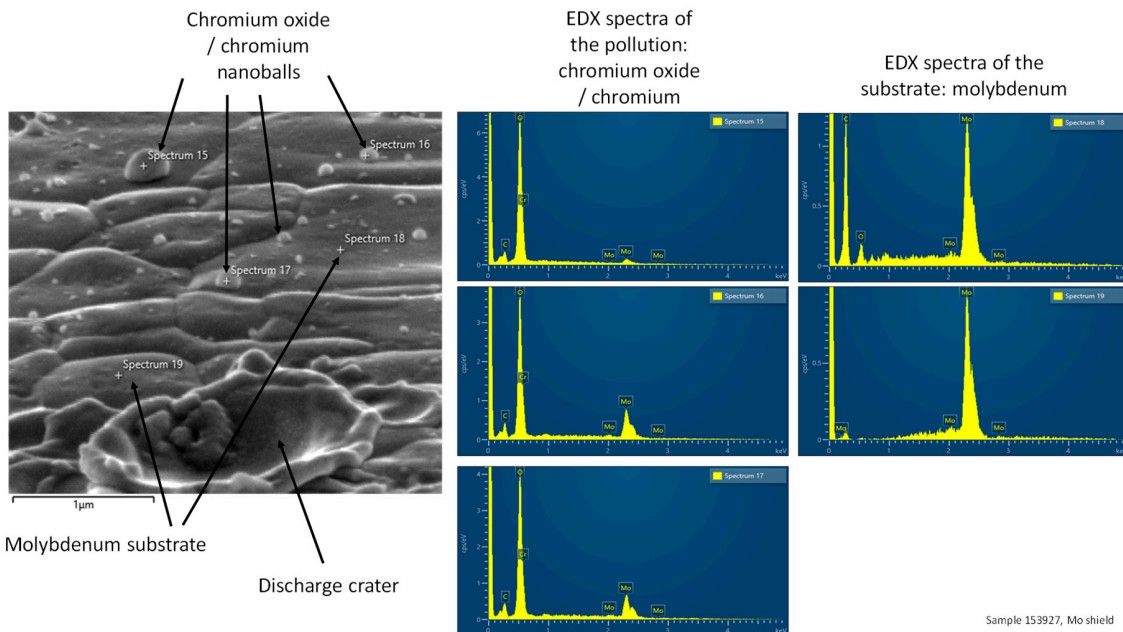

**Figure 25.** EDX spectra from CrO/Cr nanoparticles distributed across the Mo surface of a cathode ring. Oxygen is visible in spectrum 18, but missing in spectrum 19.

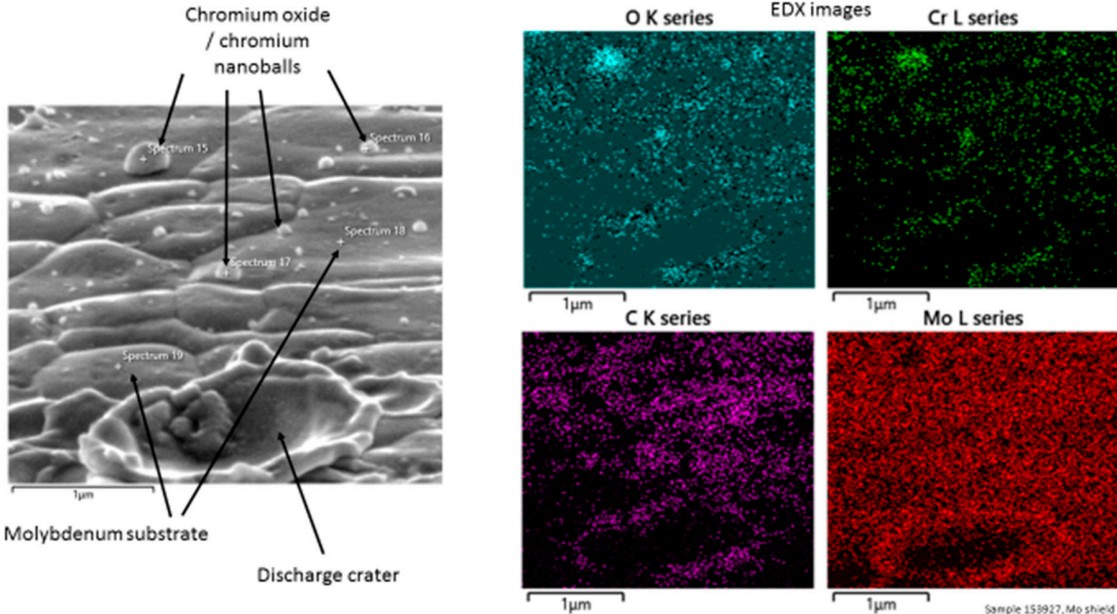

**Figure 26.** Right: Spatial distribution of the EDX signal in the left picture, from a Mo cathode ring, covered with CrO/Cr nanoparticles, which was subject to high-voltage discharge (crater at the bottom). Angulated view. There is no indication of enhanced carbon concentration in the vicinity of or underneath CrO/Cr nanoparticles.

Figure 26 shows a SEM micrograph of the same area as in Figure 25 (left) and corresponding EDX signal maps (four pictures at the right). Even in an angulated perspective, there is no sign of enhanced carbon deposition underneath any CrO/Cr nanoparticles or in their vicinity, see the map C K series. A search of other areas did not reveal clear signs of carbon fiber growth either. A single suspicious mushroom-type artifact with a length of 183 nm could be identified, which has a slightly enhanced carbon signal at the 22 nm wide stem, in comparison with its 145 nm wide cap, see Figure 27, artifact 1.

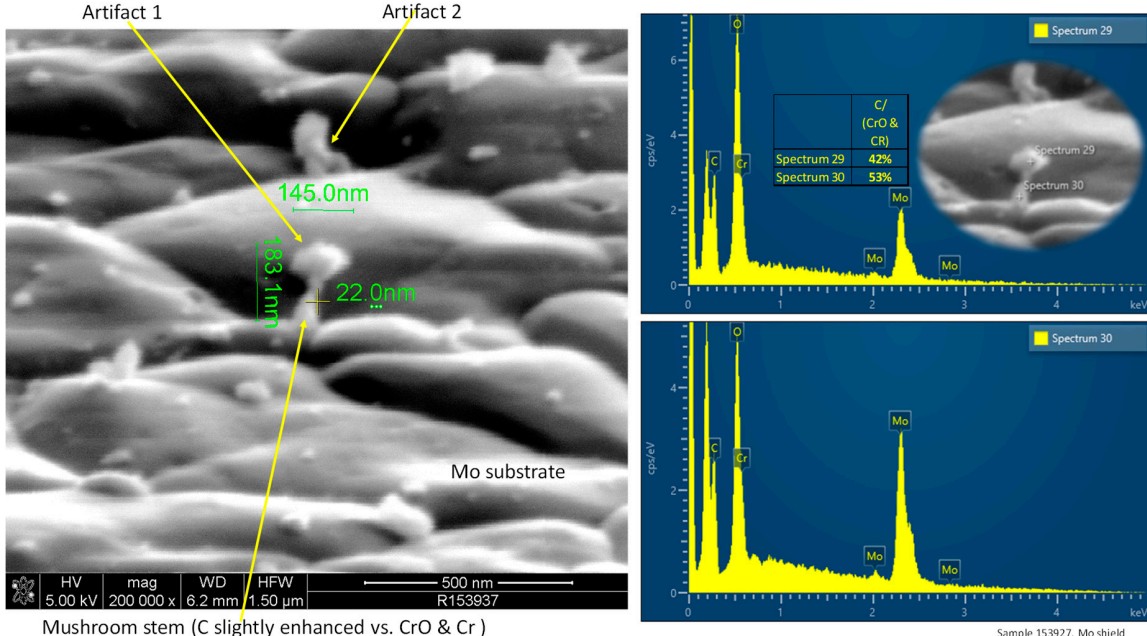

**Figure 27.** Mushroom-like artifact 1 with dimensions. The EDX signal of carbon amounts to 42% in the cap and 53% in the stem relative to the O/Cr peaks at the respective positions. Artifact 2 shows a branched stem and an elevated nanoparticle as well (no further details).

Figure 28 is a collection of other elevated artifacts found on a sample Mo ring of a cathode, see arrows.

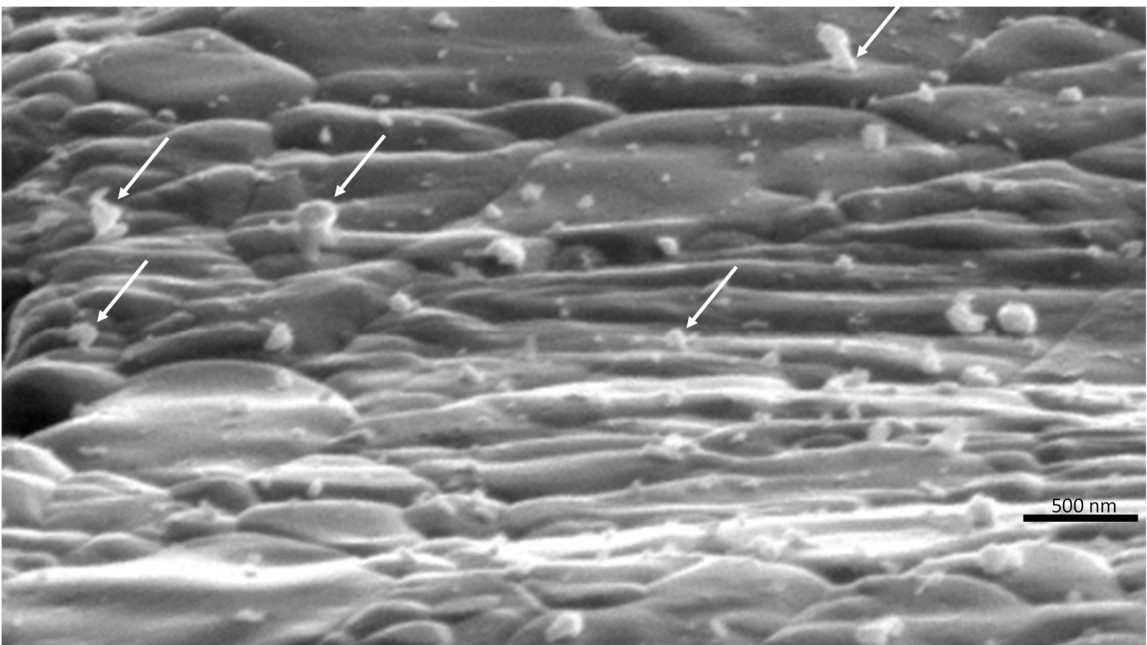

**Figure 28.** Collection of elevated artifacts (see arrows) from a sample cathode Mo ring polluted with CrO/Cr nanoparticles.

The second inspected sample Mo cathode ring showed very similar characteristics as the first one. Figure 29 depicts a characteristic image from this sample for further analysis of the cause of discharge. The artifact on the left is a copper flake of unknown origin, the right one a CrO/Cr nanoparticle, as discussed before for the other Mo ring sample. Two EDX spectra are shown on the

right, the corresponding spatial sampling areas as rectangles, overlaid to the electron image. It is obvious that for these artifacts the cross-section of the contact area with the substrate is relatively small. Modeling of EEE may consider these kind of embodiments.

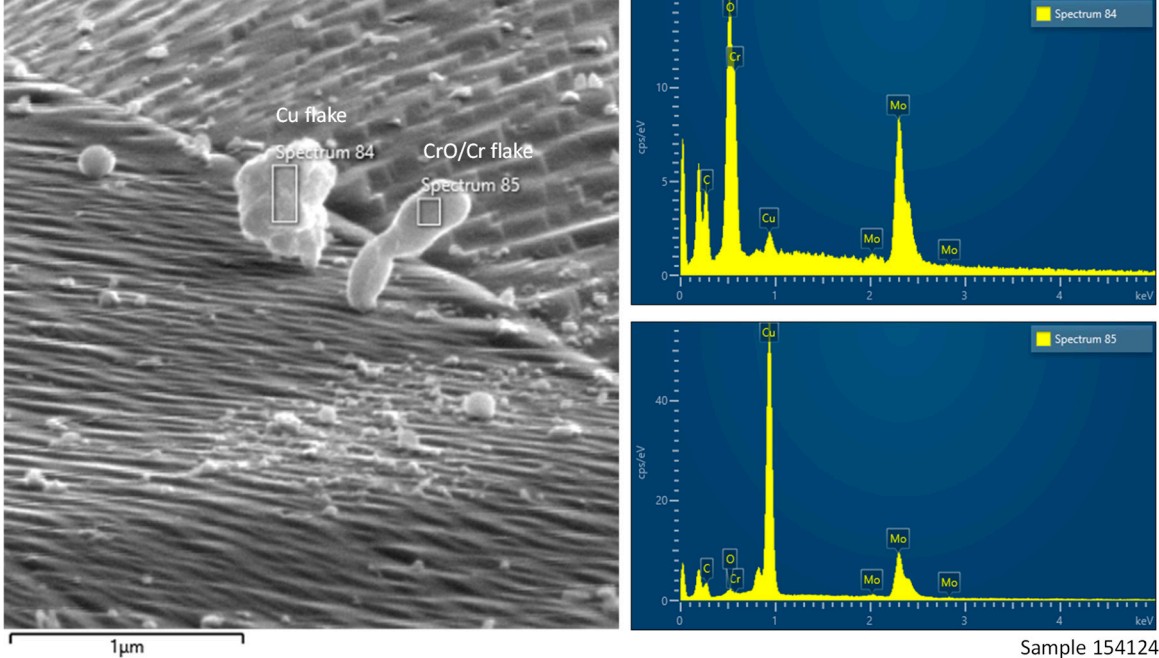

**Figure 29.** CrO/Cr (left) and copper (right) artifacts with a small contact cross-section on a Mo ring of a cathode, which was subject to high-voltage breakdown.

As carbon-based artifacts could be ruled out as the primary cause of high-voltage discharge, other aspects must be relevant. The anti-correlation of small-crater-density and chromium particle density demonstrated in Figure 24 is striking. The cathodes of the sample X-ray tubes were subject to ion bombardment with an estimated flux, as indicated in Figure 17. Ion impact, particularly by high-energy ions, may charge, heat and destruct small nanoparticles on the order of 20 nm and cause evaporation and subsequent EEE. Greaves et al. [27] report severe destruction of gold nanoclusters by 80 keV Xe ions.

*5.2. Nickel42 Cathode Head*

A third cathode from an X-ray tube type with glass frame was inspected for comparison, see Figure 30. This sample showed neither substantial pollution by nanoparticles, nor signs of carbon fiber growth. Instead, inclusions of various materials were visible at the surface, preferably accumulated along grain boundaries, with a density of about 1600/mm$^2$. The primary constituents were Si, Cl, B, N.

Figure 31 shows an exploded inclusion, which contains Cl. The cross-sectional area of the corresponding crater is about five times larger compared with the craters in its vicinity, which do not comprise residuals of an inclusion (see "spectrum 61"), and which seem to have appeared before. The distribution of discharge craters correlates with the distribution of inclusions and their appearance on crystal facets and their boundaries. See [33] for further background on electron emission from insulating particles under bombardment of charge carriers.

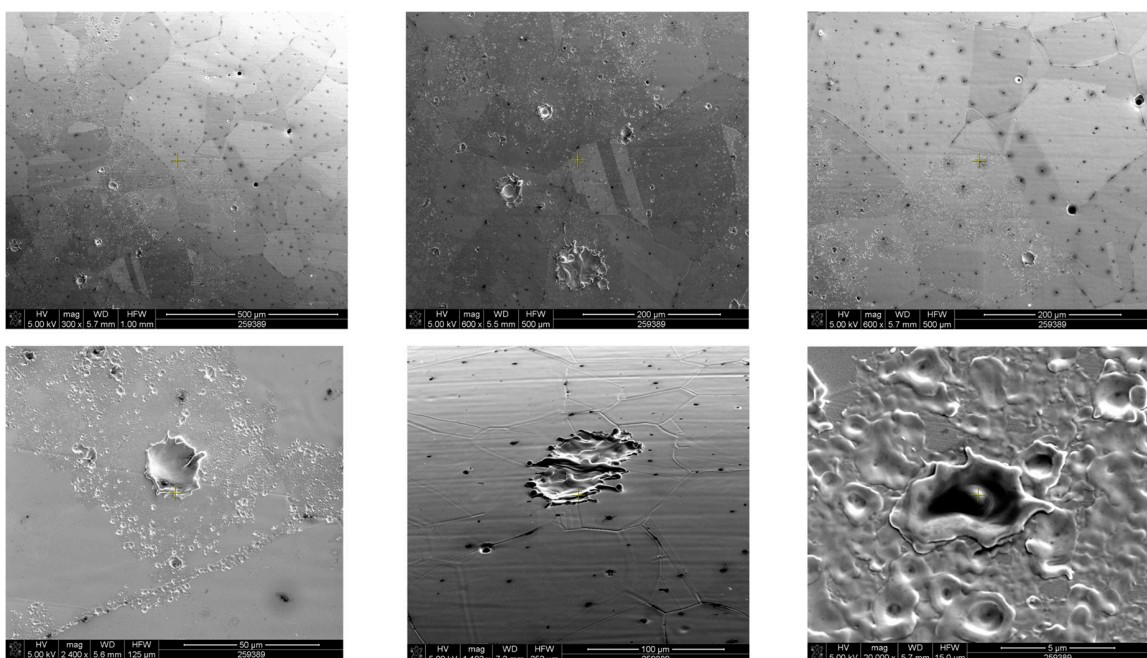

**Figure 30.** Ni42 cathode, zoom growing from top-left to bottom-right picture, which shows a characteristic discharge crater about an inclusion, which contains Cl.

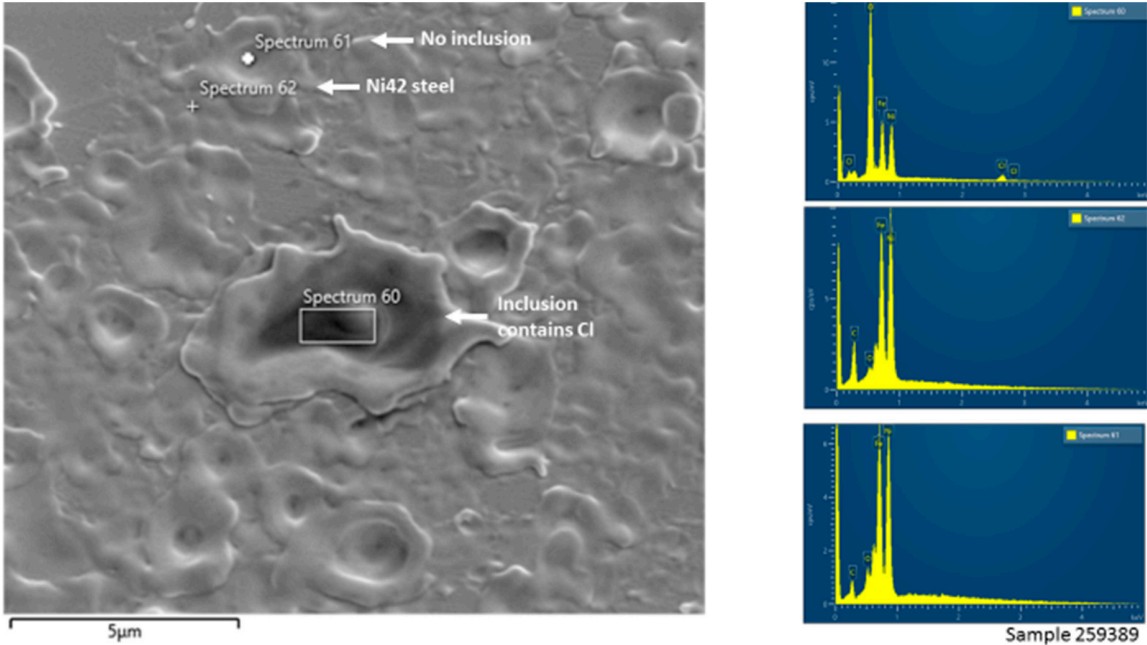

**Figure 31.** Exploded inclusion (mark "spectrum 60"), containing Cl, in the Ni42 steel cathode.

## 6. Conclusions

A new model for the emergence of carbon-based three-dimensional artifacts on electrodes under high electric fields and under ion bombardment has been proposed and discussed, which resembles growth scenarios for CNTs. Cathodes of X-ray tubes with a special population of CrO/Cr nanoparticles served as model systems. Although estimated characteristic times until breakdown fit well with the average breakdown frequency of typical medical X-ray tubes, the postulated artifacts could not be clearly identified by high-resolution SEM inspection. A single mushroom-like nanoparticle showed a

very weak excessive carbon signal at its stem. Other situations may exist, in which the proposed model may be applicable, e.g., when the vacuum system would be contaminated with carbon hydrogen to a higher degree as in a typical processed X-ray tube, the cathode temperature would exceed 800 °C, and ions would impact with kinetic energy far below 150 keV. SEM pictures indicate the significance of nanoparticle contamination and surface inclusions on breakdown behavior. Further research is advised to explain the observed correlation.

**Funding:** This research received no external funding.

**Acknowledgments:** Special thanks go to Flyura Djurabekova, Richard Forbes, Gregory Fursey, and Andreas Kyritsakis for inspiring discussions and Jack Hoppenbrouwer for taking the SEM pictures. Thanks go also to former and present colleagues of Philips, who prepared samples, photographs and ion tracing graphics, Peter Bachmann, Sarah Berhanu, Zexiang Chen, Anand Dokania, Steffen Holzapfel, Irmgard Köhler, Sven Kröger, Astrid Lewalter, Kai Lorenz, Jacqueline Merikhi, Rainer Pietig, Gereon Vogtmeier, and Tobias Wirth.

**Conflicts of Interest:** The author is employee of Philips Medical Systems DMC GmbH, Hamburg, Germany, and declares no conflict of interest.

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
