# Peer review of "Electric Field Enhancing Artifacts as Precursors for Vacuum High-Voltage Breakdown"

_instruments, doi:10.3390/instruments3040064_

Round 1

Reviewer 1 Report

The work presented in the manuscript is devoted to study of mechanism behind the sporadic electric discharge in initially conditioned vacuum devices, such as X-ray tubes. In particular, the hypothesis  of formation of Carbon nano-Tubes (CNTs) as a responsible for high Electric Field Enhancement (EFE) factors is investigated. The work includes quite inclusive review of CNT formation conditions. It is illustrated, that in principle, suitable conditions may accidentally be realized at the surface of the cathodes in realistic vacuum conditions within X-ray tubes (as an example.) The model of how the CNTs with metal blobs on their tips can trigger conditions, when Field Emission, being enhanced by EFE leads to a subsequent Explosive Electron Emission (EEE), is presented. 

However, experimental search for a signatures of CNT grown on surfaces of existing X-ray tubes, that experienced discharges, revealed no evidence of those neither hear nor at a distance from the discharge points at the cathode. Moreover, it was sown, that there exists an anticorellation between the areal frequency of discharge craters and the surface contamination with nanometric metallic particles, which could serve as seeds for formation of CNTs in question.  This strongly suppresses likelihood for the initial assumption, that the CNTs may be main reasons for the sporadic vacuum discharges, although, probably, does not completely exclude it.

The work is very interesting to be read by physicists working with similar environments, such as accelerator physics, HV engineering, particle detector physics. Even though, the latter deals with dielectric discharges in liquids and solids, underlying mechanisms of discharge initiation may be similar in some cases.

Implicitly the work suggests to search for such a discharge initiation mechanism, which would  work on a smooth uniform cathode surface, rather then on its irregularities, which is somewhat contra-intuitive, but may result in interesting findings.

The paper is very well written. There are only two remarks on the text:

page 14: Figure 15 seem to be an accidental copy of the previous one (Fig 14), while the relevant plot on range is absent;

page 15, line 365: The time, needed to saturate the seed particle with a carbon defines the initial delay before the CNT starts to form, rather than characteristic growth velocity. So I'd suggest to clarify the term "characteristic time for the growth" used by the Author to avoid ambiguity.

Author Response

Thanks for reviewing.

figure 15: corrected page 15, line 365: Text improved by differentiating saturation time and growth time and discuss dominance of the saturation time Language improved

Reviewer 2 Report

This paper presents a very interesting investigation on limitation of the reliability of a class of vacuum electronic devices due to plasma formation and breakdown between electrodes.

The electric field enhancement by conductive protuberance is a well know problem and can initiate high-voltage breakdown.

The paper proposes a new model of 3D-carbon structure grown and the experimental investigation of this effect on three sets of electrodes. The experimental arrangement and surfaces inspection are described in detail. The need of further research to fully understand the laboratory observations is  correctly suggested by the author in the conclusion.

The paper is very interesting, well written and gives a coherent account of the research undertaken.

Author Response

Thank you for the revision and comments

I improved the text, implemented minor corrections and the language.
